# Wildfire smoke-plume rise: a simple energy balance parameterization

Nadya Moisseeva[1] and Roland Stull[1]

[1]Department of Earth, Ocean and Atmospheric Sciences, The University of British Columbia, Vancouver BC V6T 1Z4, Canada

**Correspondence:** Nadya Moisseeva (nmoisseeva@eoas.ubc.ca)

**Abstract.** The buoyant rise and the resultant vertical distribution of wildfire smoke in the atmosphere have a strong influence on downwind pollutant concentrations at the surface. The amount of smoke injected vs. height is a key input into chemical transport models and smoke modelling frameworks. Due to scarcity of model evaluation data as well as inherent complexity of wildfire plume dynamics, smoke injection height predictions have large uncertainties. In this work we use a coupled fire-atmosphere model WRF-SFIRE configured in large eddy simulation (LES) mode to develop a synthetic plume dataset. Using this numerical data, we demonstrate that crosswind integrated smoke injection height for a fire of arbitrary shape and intensity can be modelled with a simple energy balance. We introduce two forms of updraft velocity scales that exhibit a linear dimensionless relationship with the plume vertical penetration distance through daytime convective boundary layers. Lastly, we use LES and prescribed burn data to constrain and evaluate the model. Our results suggest that the proposed simple parameterization of mean plume rise as a function of vertical velocity scale offers reasonable accuracy (30 m errors) at little computational cost.

## 1 Introduction

Predictions of surface concentrations of wildfire smoke by regional and global chemical transport models depends on the initial equilibrium height of the smoke plume. Plume rise, which determines this equilibrium height, is widely recognized as an area of uncertainty (Goodrick et al., 2013; Paugam et al., 2016). Traditionally, many operational smoke modelling frameworks relied on plume rise equations originally developed by Briggs (1975) for industrial smokestacks (Larkin et al., 2010; Pavlovic et al., 2016). Yet several studies suggest that this approach may not be appropriate for wildfires (Pavlovic et al., 2016; Heilman et al., 2014; Freitas et al., 2007).

In a recent review of existing plume rise parameterizations, Paugam et al. (2016) highlight three notable models that stand out in literature, as that of Freitas et al. (2007), Sofiev et al. (2012) and Rio et al. (2010). Both Freitas and Rio's approaches use first principles to characterize plume temperature, vertical velocity and entrainment. While the former provides prognostic 1-D equations that can be solved as a stand-alone "offline" model, the latter is implemented as a sub-grid effect within a host chemistry transport model. Notably, both consider an idealized heat source to represent the fire. To initialize the plume at the lower boundary, simplified fire geometry (circular and rectangular for Freita's and Rio's models, respectively) with a

uniform heat flux is assumed. Sofiev's semi-empirical approach relies on energy balance and dimensional analysis, while using satellite data to both initialize and constrain the parameterization. Unlike Briggs's equations, all of the above models address wildfire plumes specifically, yet much research is needed to reduce the large uncertainties associated with the model predictions (Mallia et al., 2018). Moreover, it is unclear, whether unreliable predictions should be attributed to the fire input parameters or the plume rise model itself.

One of the central challenges in plume rise model development has been the scarcity of comprehensive model evaluation data (Coen et al., 2012b; Ottmar et al., 2016). To date, information on wildfire smoke emissions and dispersion has largely been derived from two distinct sources: remotely sensed data and prescribed burn campaigns. While increasing numbers of satellite observations contribute to a more complete plume climatology (Val Martin et al., 2010), the data is subject to biases and lacks direct spatiotemporal links to fire behavior (Ichoku et al., 2012). In contrast, field campaigns, such as Prescribed Fire

Combustion and Atmospheric Dynamics Research Experiment (RxCADRE) (Ottmar et al., 2016) and Fire and Smoke Model Evaluation Experiment (FASMEE) (Prichard et al., 2019), provide the necessary level of detail for model validation studies. However, such datasets typically capture a modest range of fire and atmospheric conditions.

Our approach, therefore, is to develop a synthetic plume dataset that addresses the limitations of the available observational data. As vast majority of smoke plumes remain in or just above the atmospheric boundary layer (ABL) (Val Martin et al.,

2010; Mallia et al., 2018), we use large eddy simulations (LES) to focus on local- and meso- scale plume dynamics. Using a coupled model, we simulate a wide range of fire and atmospheric conditions (Sect. 2). Based on this synthetic LES data (hereafter referred to as "data") we propose a simple energy balance model for predicting plume rise of crosswind integrated (CWI) smoke from a non-uniform fireline (Sect. 3). We use the synthetic plume dataset to constrain and evaluate our plume rise parameterization. We then demonstrate with both numerical and prescribed burn data, that within the range of tested conditions

this parameterization offers high speed and accuracy (Sect. 4). Moreover, it provides the means for classifying penetrative vs. non-penetrative plumes, which is key for subsequent dispersion modelling (Sofiev et al., 2012; Val Martin et al., 2012).

The proposed approach is geared toward regional smoke modelling frameworks (e.g. BlueSky and BlueSky Canada). Government agencies, air quality managers and fire response teams depend on these operational tools and their accuracy to issue air quality warnings, evacuation orders and to help mitigate human health impacts. Yet, model evaluation studies suggest that

plume rise estimation remains a weak link within smoke modelling systems (Raffuse et al., 2012; Val Martin et al., 2012; Chen et al., 2019). Moreover, existing methods struggle to reliably differentiate which plumes remain in the ABL and which penetrate it. The broad goal of the work is, therefore, to address some of these challenges and improve the accuracy of plume rise predictions for regional air quality applications.

## 2  Development of a Synthetic Plume Dataset

We devise a synthetic plume data set using a coupled fire-atmosphere model WRF-SFIRE, which combines the well-established Weather and Research Forecasting Model (WRF) with a semi-emperical fire spread algorithm called SFIRE (Mandel et al., 2014; Mallia et al., 2020; Coen et al., 2012a; Kochanski et al., 2013; Clements et al., 2006; Kochanski et al., 2019). The model

allows one to explicitly resolve plume dynamics, while parameterizing fuel combustion. One of the primary advantages of using WRF-SFIRE is that it supports two-way coupling between the atmosphere and the fire behavior model, allowing it to capture some of the complex dynamical feedbacks that exist between the fire and the atmosphere (Prichard et al., 2019). Heat and moisture fluxes from the simulated burn provide forcing to the atmosphere, affecting local wind flow and thermodynamics. This in turn influences the modelled fire behavior. The following sections detail the numerical setup of WRF-SFIRE, scope of the dataset, as well as our approach to defining "ground truth" for model evaluation.

## 2.1 Numerical Configuration

WRF-SFIRE was configured in idealized large-eddy resolving mode. Much of our numerical setup was adopted from a case study of a real prescribed burn as detailed in Moisseeva and Stull (2019), to ensure the simulations represent physical conditions backed by model evaluation. Due to high computational demands of LES runs, we focused on the local- and meso- scales, considering only the initial buoyant plume rise of smoke in typical daytime atmospheres. Key parameters varied were ambient wind, fuel category, vertical potential temperature profile and fireline length, denoted as conditions **W**,**F**,**R** and **L**, respectively (detailed further in Sect. 2.2).

Each 10 km x 20 km domain with 40 m horizontal grid spacing was initialized with uniform ambient west wind **W** and vertical temperature profile **R**. Depending on the sounding **R**, the simulations were performed in either a shallow (3000 m) or a deep (5000 m) domain, with 51 or 71 hyperbolically stretched vertical levels, respectively. A constant uniform lower boundary surface thermal flux (tke_heat_flux) in the ambient environment and lateral periodic boundary conditions were imposed to produce a turbulent well-mixed layer. We used full surface initialization (sfc_full_init =.true.), with the lower boundary characteristics set to USGS values for land use most closely matching the Anderson fuel category **F** (Anderson, 1982). The corresponding surface roughness lengths added various levels of wind shear to each domain to produce a more realistic non-uniform vertical wind profile during spinup of the environment before the fire was initialized in the LES.

Initial convection in the ambient environment was triggered using a perturbed surface temperature field. On average, a near-stationary turbulence spectrum was achieved within the first 30 min of run start. The "restart" file generated at the end of one hour of spinup was used to initialize the main burn simulation, ensuring the fire was ignited in a well-mixed turbulent ABL.

The fire was initialized over a one-minute interval using a straight line of length **L**. The ignition line was placed one kilometer downwind of the western edge of the domain and centered in the north-south direction (sample illustration of this setup can be found in Appendix A). With a refinement ratio of 10 in each horizontal direction, the fire was simulated on a 4 m sub-grid mesh.

The "smoke plume" was modelled with a passive tracer emitted proportionally to the mass and type of fuel burned. The rate of release was controlled by an assigned emission factor representing $PM_{2.5}$ for each fuel category, based on values provided by Prichard et al. (2017) (see namelist.fire_emissions in Supplementary Material).

A summary of key configuration details can be found in Table 1, as well as in sample namelist initialization files provided as Supplementary Material.

**Table 1.** Key parameters of numerical domain setup.

| Simulation Parameter | Value/Description |
|---|---|
| Model version | May 24, 2019 (git #ced5955) |
| Horizontal grid spacing | 40 m |
| Domain size | 500 grids cells (east-west) x 250 grids cells (north-south) |
| Time step | 0.1 s |
| Model top | 3000 m (shallow) / 5000 m (deep) |
| Spinup timing | 11:30:00 - 12:30:00 |
| Fire (restart) simulation timing | 12:30:00 - 12:50:00 (shallow) / 12:30:00 - 13:00:00 (deep) |
| Sub-grid scale closure | 1.5 TKE (TKE = Turbulence kinetic energy) |
| Lateral boundary conditions | periodic |
| Surface physics | Monin-Obukhov similarity (sf_sfclay_physics = 1) |
| Land surface model | thermal diffusion (sf_surface_physics = 1) |
| Ambient surface heat flux | 240 $\mathrm{Wm}^{-2}$ (tke_heat_flux=0.2) |
| Fire mesh refinement | 10 |
| Ignition duration | 13:00:10 – 13:01:10 |
| Heat of combustion of dry fuel | 16.4e+06 J $\mathrm{kg}^{-1}$ |

**Table 2.** Test conditions included in synthetic plume dataset. The count indicates the number of unique values used within the specified range.

| Condition (Tag) | Range | Count | Description |
|---|---|---|---|
| Ambient wind (W) | 3 - 12 $\mathrm{ms}^{-1}$ | 10 | Uniform horizontal wind magnitude used to initialize model spinup |
| Stability profile (R) | R0-R8 | 9 | Atmospheric sounding with variable ABL height, temperature and inversion strength |
| Fuel (F) | 1 - 13 | 13 | Anderson fuel category assigned at lower boundary |
| Fireline length (L) | 1 - 4 km | 3 | Length of ignition line |
| | | | **Total number of experiments = 140** |

## 2.2 Test Conditions

Table 2 summarizes the key parameters that were varied to produce the synthetic dataset. The reason for considering the given conditions is twofold: these parameters (i) have been widely acknowledged as having a strong impact on plume behavior and (ii) can be obtained (and provided as input for the parameterization) under real-world scenarios.

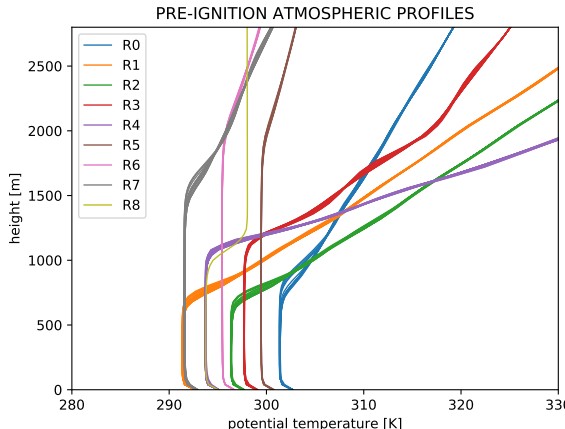

**Figure 1.** Pre-ignition potential temperature profiles (stability condition **R**). Colors correspond to initial soundings used for model spinup.

The range of ambient winds tested was bound largely by numerical constraints. Due to cyclic boundary conditions, wind speeds higher than $12 \text{ ms}^{-1}$ would require a much larger domain to prevent smoke recirculation. For the lower bound on our wind condition **W**, we needed to ensure that sufficient wind speed was maintained to propagate the fire. The spread algorithm used within the LES applies a correction factor under low wind speed conditions to prevent the fire from extinguishing itself. While necessary for numerical reasons this effect is not physical, so winds below $3 \text{ ms}^{-1}$ were excluded from our dataset.

We used 9 different atmospheric profiles (**R** condition) to initialize the model. We varied the following features for each initialization:

- initial ABL height ($500 \text{ m}$ - $1600 \text{ m}$)

- potential temperature lapse rate above inversion ($0 \text{ K km}^{-1}$ - $20 \text{ K km}^{-1}$)

- initial ABL temperature ($290 \text{ K}$ - $300 \text{ K}$)

Following spinup (Sect. 2.1) under variable winds and surface conditions, this produced 9 sets of soundings, shown in Fig. 1 with ABL depths of approximately $600 \text{ m}$ - $2000 \text{ m}$.

     We tested all fuel categories available within the model (**F** condition), and varied the length of the fireline (**L** condition) between 1 and 4 km. Weakly buoyant non-penetrative plumes whose smoke remained within the well-mixed ABL were excluded from the dataset, as their behavior is governed by different physics. A tabulated summary of all combinations included can be 110 found in Appendix B.

     Note, that varying a single condition while holding the rest constant does not result in a controlled experiment isolating its impact on plume rise. Because WRF-SFIRE incorporates fire-atmosphere coupling, the problem is not well-constrained. For example, by varying fuel type **F** alone, while holding the rest of test conditions constant, we obtain a set of fires with diverse shapes, sizes, intensities, fireline depths, rates of spread and heat release. This reflects the complexity of non-linear interactions

that exist between the fire and the atmosphere. As a result, the parameter space captured within our LES dataset is much greater then the four conditions described in Table 2.

## 2.3 Defining Smoke Injection Height

Given non-stationary fire and atmospheric conditions, determining a consistent definition of an equilibrium smoke injection height is not a trivial task. It requires separating buoyant rise from dispersion, while excluding the effects of initial momentum
overshoot and accounting for the advection due to varying ambient and fire-generated winds.

A common way of examining vertical distributions of pollutants in the context of air quality is to consider CWI concentrations. This allows to reduce the problem to two dimensions, with plume centerline being defined simply as the CWI concentration maximum at each location downwind of the source. Theoretically, under stationary conditions there exists an equilibrium height, around which the centerline eventually oscillates. In reality, as well as in our LES experiments, neither the ambient nor
the fire conditions are stationary. The changing location, shape and intensity of the fire, ABL warming and growth, as well as the development of fire-coupled winds and vorticity continually modify the conditions.

As a result, our approach is based on defining a region, where the concentration distribution is quasi-stationary. We consider the last frame of each simulation for this analysis. Using CWI integrated tracer values, we locate the plume centerline (Fig. 2a). Due to random effects of ABL thermals as well as fluctuations in fire intensity and propagation speed, both centerline
height and concentration can vary near the heat source. These oscillations are naturally suppressed in the stable layers above the ABL, as the plume travels downwind and undergoes additional widening and mixing. To obtain the quasi-stationary region for each individual plume, we first calculate the change in tracer concentration along the centerline. We then use a smoothing function to reduce the effect of random turbulent oscillations in both the centerline height and the tracer concentration gradient along the centerline. The downwind region where both of these parameters are not changing rapidly are then then considered
quasi-stationary. Additional details of this filtering method are provided in Appendix C.

The vertical CWI distribution of tracers are then averaged in the downwind direction over the identified quasi-stationary regions (shaded in grey on the Fig. 2c) to produce a representative downwind distribution for each plume (Fig. 2d). We define the "true" injection height $z_{CL}$ as the mean height of smoothed centerline over the averaging region. The resultant dataset of $z_{CL}$ values is used to constrain and evaluate the proposed smoke injection height parameterization introduced in the following
sections.

## 3 Smoke Injection Height Model for Penetrative Wildfire Plumes

A common approach to predicting the final equilibrium centerline height of wildfire smoke is to first estimate the initial buoyant energy of the hot rising smoke (Briggs, 1975; Sofiev et al., 2012; Anderson et al., 2011). After the smoke plume entrains surrounding ABL environmental air and cools, the remaining energy is spent doing work to push the cooled smoke
plume up into the statically stable capping inversion.

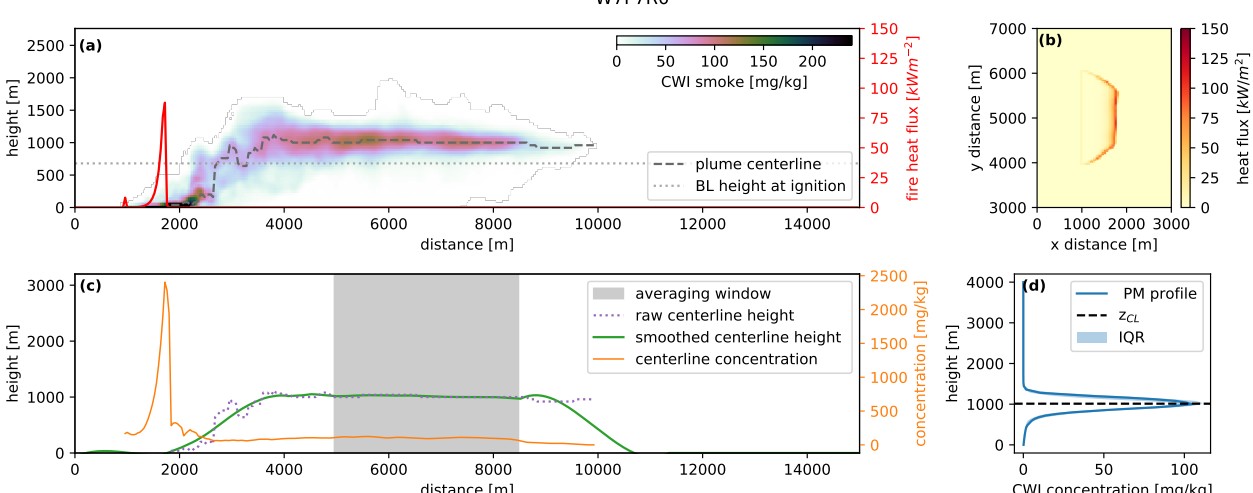

**Figure 2.** Illustration of the approach to identifying a quasi-stationary downwind region in CWI smoke distribution using a sample LES experiment. (**a**) CWI smoke concentrations. Also shown are plume centerline height (dashed), $z_i$ (dotted) and CWI fireline intensity (solid red, secondary axis). (**b**) Plan view of fire heat flux showing the fireline. (**c**) Quasi-stationary region (grey shading). Also shown are raw (dotted purple) and smoothed (solid green) centerline heights and the tracer concentration gradient (solid orange, secondary axis). (**d**) Representative downwind smoke distribution. The profile (solid blue line) is obtained by horizontally averaging the CWI smoke concentrations in the quasi-stationary region (dashed grey in (c)). Also shown are IQR (light blue shading) and the derived smoke injection centerline height $z_{CL}$ (dashed black).

The relationship between final and initial energies is often rewritten to show that the potential energy per unit mass (PE) of smoke penetration equals some fraction $c_1$ of initial heat released from the fire. In kinematic units, the initial heat input has units similar to a kinetic energy per unit mass (KE). The empirical parameter $c_1$ is usually estimated based on concepts of entrainment into the rising smoke plume (Cushman-Roisin, 2014).

$$PE = c_1 KE \tag{1}$$

The PE of smoke-plume penetration into the capping inversion can be written as

$$PE = g'z' \tag{2}$$

where the penetration distance $z'$ of the final equilibrium smoke centerline $z_{CL}$ above reference height $z_s$ (near the top of the well-mixed portion of ABL) is

$$z' = z_{CL} - z_s \tag{3}$$

The static-stability variable $g'$ for the plume-penetration region is

$$g' = g\frac{\theta_{CL} - \theta_s}{\theta_s} = g\frac{\theta'}{\theta_s} \tag{4}$$

where $\theta_{CL}$ and $\theta_s$ are the potential temperatures of the ambient environment at $z_{CL}$ and $z_s$, respectively, and $\theta_{CL} - \theta_s = \theta'$.

The KE can be estimated using a velocity scale $w_f$ as

$$KE = 0.5 w_f^2 \tag{5}$$

Traditionally, the bulk potential-temperature difference across the smoke-plume penetration region $\theta'$ is expected to be relevant for only the PE portion of Eq. (1). However, we found from the LES runs for a wide range of fire and environment conditions that the KE also depends on the same potential temperature difference. This dependence can be expressed in the velocity scale:

$$w_f = \frac{I}{z_i \theta'} \tag{6}$$

This velocity scale is related to the fireline intensity parameter $I$, which is the kinematic heat flux into the atmosphere integrated across the fireline depth (in units of $\mathrm{Km^2 s^{-1}}$), and to the mixed-layer depth $z_i$. Note, that $I$ effectively corresponds to the kinematic form of Byram's Fireline Intensity (in units of $\mathrm{kWm^{-1}}$).

We speculate that this interesting result is because smoke from a fire does not rise through a passive environment, as is often assumed for Briggs types of plume entrainment models. Instead, the fire and the environment interact in many complex ways. Some of these include: vertical-to-bent-over vortices on the ends of the fire line that rapidly mix environmental air into the buoyant smoke plume; modulation of fire intensity and fire updrafts by translation of ambient thermals across the fire line; plumes of enhanced convergence and updraft along the fire line; mass conservation as descending air beneath the extended smoke plume lowers the local mixed-layer depth; and other factors.

Thus, Eq. (1) becomes

$$g' z' = c_2 \left[ \frac{I}{z_i \theta'} \right]^2 \tag{7}$$

where $c_2 = 0.5 c_1$.

The above can be rearranged into the following form

$$z_{CL} - z_s = C \left[ \frac{g \left( \theta_{CL} - \theta_s \right)}{\theta_s \left( z_{CL} - z_s \right)} \right]^{-\frac{1}{2}} \left\{ \frac{g I \left( z_{CL} - z_s \right)}{\theta_s z_i} \right\}^{\frac{1}{3}} \tag{8}$$

where the dimensionless empirical parameter is $C \approx 1$. The factors in square and curly brackets with their corresponding powers have units of time and velocity, respectively. This relationship is plotted in Fig. 3. It provides quite an acceptable fit to the data over a wide range of 140 combinations of fire and atmospheric conditions simulated.

Equation (8) suggests that the relevant length and temperature scales $(z', \theta')$ depend not on the capping inversion strength alone, or on the tropospheric lapse rate above the capping inversion alone, but on the bulk potential-temperature differences across the smoke-plume penetration region, $z_{CL} - z_s$. Eq. (8) is implicit, in that the desired plume centerline equilibrium height $z_{CL}$ appears in both the left and right sides of the equation. The plume centerline height also defines where $\theta_{CL}$ is retrieved from the atmospheric sounding; namely, $z_{CL}$ is implicit in both Eq. (7) and (8). However, for any specific fire and environment

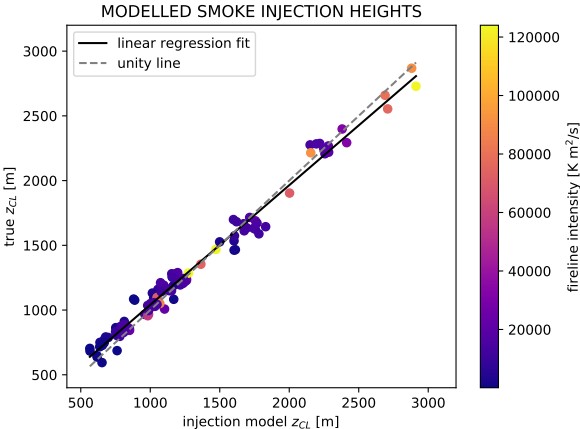

**Figure 3.** Comparison of true (as shown in Fig. 2) and modelled (from Eq. (8)) smoke injection heights. Scatter points represent the 140 individual plume experiments within the LES dataset, with colors corresponding to fireline intensity $I$. Solid black and grey dashed lines denote linear regression fit and unity, respectively.

conditions, values of $z_{CL}$ are easily found by iteration (see Appendix E). Steps to estimating input parameters required for the proposed injection model from the LES data are summarized in Appendix D.

Alternatively, for a small sacrifice in accuracy, we can obtain an explicit solution by considering an idealized version of the atmospheric profile, consisting of an adiabatic mixed layer, entrainment zone and a stable uniformly stratified free atmosphere above (Fig. 4). In such case $\gamma$ is defined as the overall potential temperature gradient of the free atmosphere and $z_s$ as the height corresponding to the intercept of $\gamma$ and the well mixed portion of the ABL profile. Then, using Eq. (8), $z_{CL}$ can be found explicitly as:

$$z_{CL} = C^{\frac{3}{2}} \left[ \frac{\theta_s}{g} \right]^{\frac{1}{4}} \left[ \frac{I}{z_i} \right]^{\frac{1}{2}} \left[ \frac{1}{\gamma} \right]^{\frac{3}{4}} + z_s \qquad (9)$$

## 4   Results

To assess the accuracy of the proposed smoke injection height parameterization (Eq. (8)), we performed two sets of verification studies. The first approach is based on using the synthetic plume dataset to perform model evaluation, bias correction and sensitivity analysis with idealized data. The second portion of this section applies our approach to a case study of a real

prescribed burn (RxCADRE 2012).

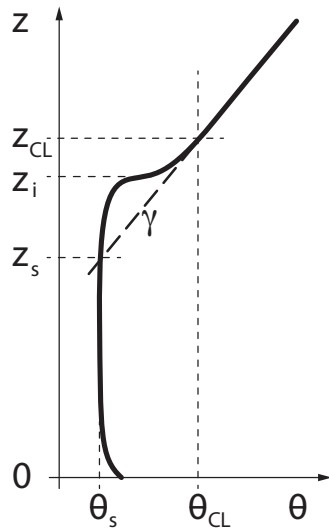

**Figure 4.** Idealized potential temperature profile $\theta$ vs. height with constant stable layer lapse rate $\gamma$.

## 4.1 Numerical Results

Shown in Fig. 3 are "true" and parameterized smoke injection heights. The former is obtained directly from the LES, as per Sect. 2.3. The latter is determined iteratively using the proposed smoke injection height parameterization (see Appendix E for implementation details).

Individual prediction errors do not appear to be a function fireline intensity, as indicated by scatter point color in Fig. 3, or ambient winds (not shown). While overall the model performance is encouraging, the small discrepancy between the unity and regression lines suggests a linear bias. This can be remedied by applying bias correction using regression parameters from the fit shown in Fig. 3. This optimized model produces errors on the order of 20 - 30 m, as suggested by the interquartile range shown in Fig. 5d. Model bias will be addressed in further detail in Sect. 5.

Given smooth averaged profiles from the synthetic dataset and excluding condition **R8** (adiabatic free atmosphere), the explicit solution using Eq. (9) offers comparable accuracy to the iterative version for both raw and bias corrected datasets (Fig. 6) . We address the limitations of using the explicit approach in Sect. 5.5.

## 4.2 Model Sensitivity

To asses how sensitive the smoke injection model performance is to the particular choice of bias correction parameters, we
partition our original plume dataset into training and testing groups through random sampling. We obtain the linear bias correction parameters using training data only (80% of runs). We then apply our bias-corrected iterative solution to the test group (remaining 20% of the runs) and assess model accuracy. Figure 7 summarizes model performance and sensitivity, based

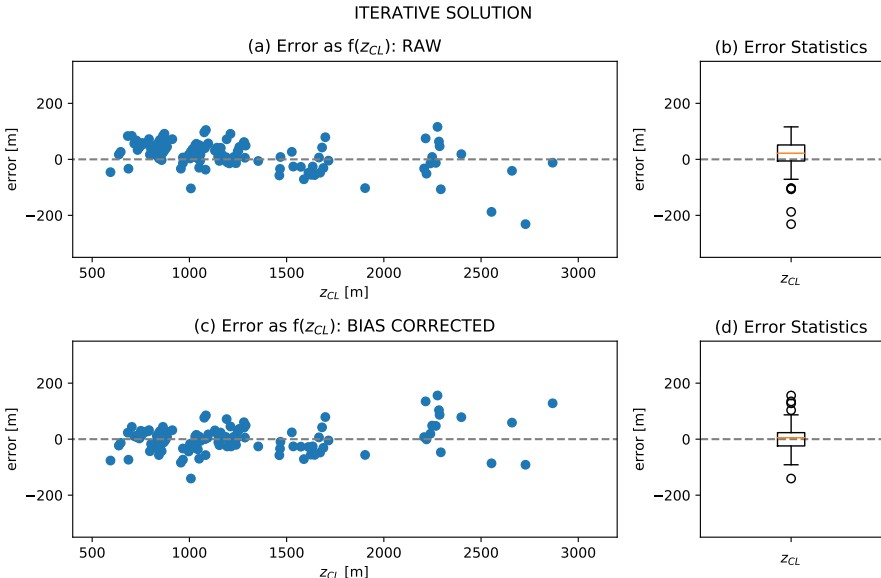

**Figure 5.** Performance of the smoke injection height parameterization based on the iterative solution (Eq. (8)). (a) Non-bias corrected model prediction error (true - modelled $z_{CL}$) as a function of $z_{CL}$. (b) Error statistics for non-bias corrected model. The box and whiskers span interquartile range (IQR) and 1.5 x IQR, respectively. Median value shown in orange. (c) Bias-corrected model prediction error as a function of $z_{CL}$. (d) Error statistics for bias-corrected model.

on 10 trials of sampling with replacement. Consistently high Pearson correlation shown in the trial histogram in Fig. 7c, are encouraging, and suggest that the particular choice of simulations used in bias correction does not have a strong impact on model accuracy.

### 4.3 Evaluation with Observations

Next, we apply the proposed model to a real-life case-study. We use observational data from the RxCADRE L2G prescribed burn (Ottmar et al., 2016) and it's numerical simulation (Moisseeva and Stull, 2019). This case study was selected based on (i) comprehensive nature of the observational dataset (ii) penetration of the plume above ABL top and (iii) availability of a completed model validation study, confirming that WRF-SFIRE reasonably captures the smoke plume produced during the burn.

Shown in Fig. 8 is the strip headfire pattern used to ignite the grass plot. We estimate the burn's input fireline intensity parameter $I$ in two different ways: from raw data collected during the burn as well as from the numerical simulation.

The observations-based value $I_{obs}$ is derived from the integral heat flux data obtained from the Highly Instrumented Plots (HIPs) fire behavior package (FBP) sensors (Jimenez and Butler, 2016). We use the provided time-integrated values, averaging between all sensors with confirmed fire at the sensor location (as indicated by video footage (Butler et al., 2016)). We then

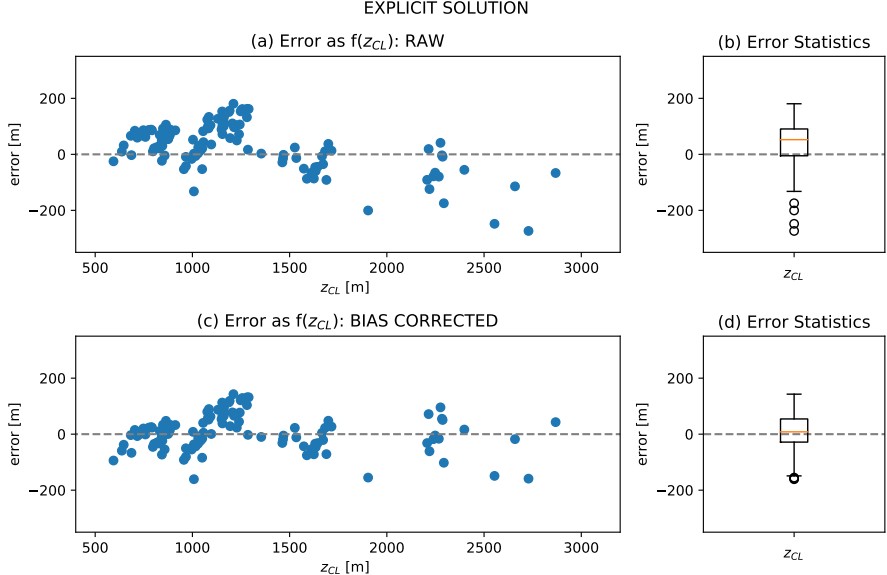

**Figure 6.** Performance of the smoke injection height parameterization based on the explicit solution (Eq. (9)). (a) Non-bias corrected model prediction error (true - modelled $z_{CL}$ ) as a function of $z_{CL}$. (b) Error statistics for non-bias corrected model. The box and whiskers span interquartile range (IQR) and 1.5 x IQR, respectively. Median value shown in orange. (c) Bias-corrected model prediction error as a function of $z_{CL}$. (d) Error statistics for bias-corrected model.

obtain the mean value (in kinematic units) of $236\ \mathrm{Kms}^{-2}$ and multiply it by the average measured rate of spread (ROS) of 0.38 $\mathrm{ms}^{-1}$ (Butler et al., 2016) for the same sensors to convert to spatially-integrated heat flux for a single fire line. We assume that this value is representative of the remaining three firelines, hence:

$$I_{obs} = 236 \cdot 0.38 \cdot 4 = 359 \tag{10}$$


in units of $\mathrm{Km}^2\mathrm{s}^{-1}$. Note, that raw data for both heat fluxes and ROS values have extremely large associated uncertainties. Observed ROS values vary by nearly a factor of two, depending on the measurement technique used. While we have included only locations with ignition confirmed by video footage in our calculations, heat fluxes still vary up to a factor of four between sensors.

For comparison, we also obtain an LES-based integrated fireline intensity value $I_{LES}$. Due to wind shear, as measured by the sounding launched prior to the burn (10:00:00 CST), the CWI direction at the surface differs from the one used to estimate CWI smoke. $I_{LES}$ was, hence, estimated by assuming 125 degree rotation of LES fields, based on the lowest available wind direction measurement. We use trapezoidal rule to numerically integrate the mean crosswind heat flux along the depth of the fireline (see Appendix D) and find $I_{LES} = 1002\ \mathrm{Km}^2\mathrm{s}^{-1}$.

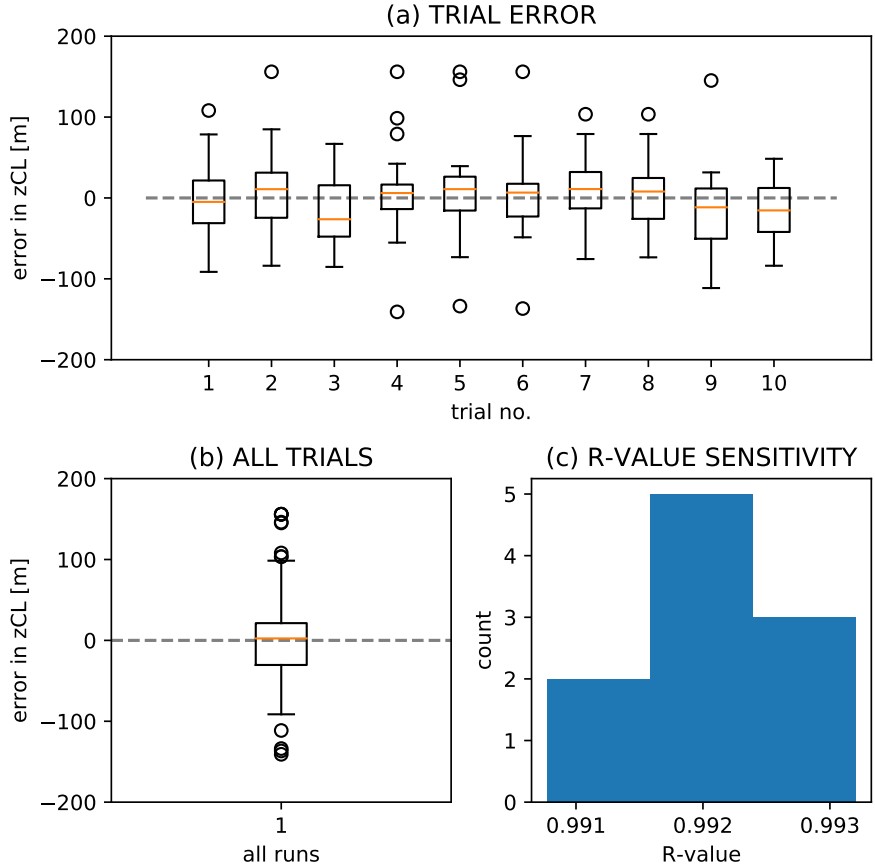

**Figure 7.** Analysis of model sensitivity to the choice of bias correction parameters. (a) Error distributions for individual trials using independent (test) data. (b) Error distribution for all trials using independent (test) data. (c) Sensitivity of R-value (correlation coefficient) for all trials.

We apply our iterative solution (Eq. (8)) to find two $z_{CL}$ estimates based on $I_{obs}$ and $I_{LES}$, and compare them to the CWI smoke injection height obtained from the LES. The results are shown in Fig. 9. The parameterized injection heights are under-predicted by 20 m and 70 m for LES- and observations- derived $I$ values, respectively.

## 5 Discussion

### 5.1 Plume Classification

In previous sections we apply an energy balance parameterization to predict the mean smoke injection height $z_{CL}$ of a given penetrative plume. For this purpose, only plumes rising above ABL top $z_i$ were included in the synthetic plume dataset used

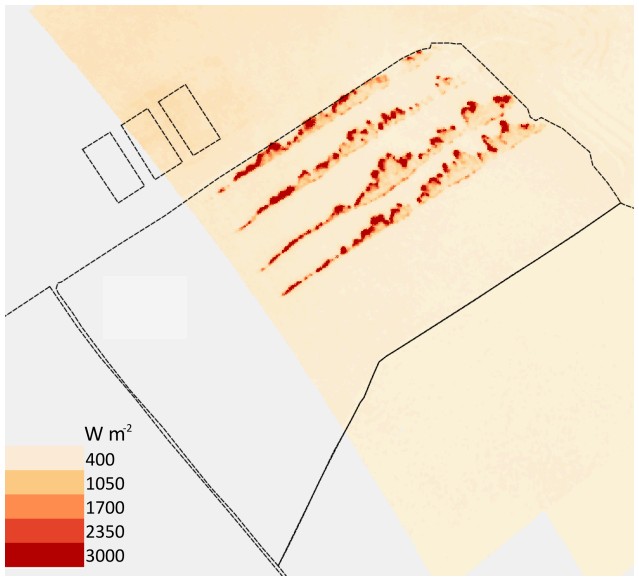

**Figure 8.** Long wave infra-red (LWIR) image of L2G lot during ignition (12:32:02 CST) with dashed black lines denoting burn perimeters.

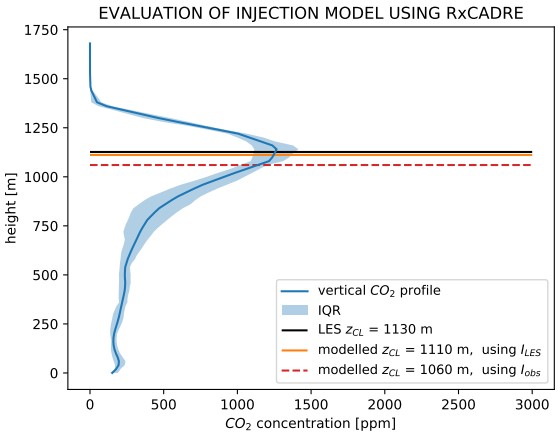

**Figure 9.** Model evaluation using a case-study of a real prescribed burn (RxCADRE 2012). CWI smoke concentration profile shown in blue. "True" $z_{CL}$ obtained directly from LES shown in solid black. Solid orange and dashed red lines correspond to $z_{CL}$ estimates obtained using the iterative solution of the proposed smoke injection height parameterization (Eq. (8)), based on LES- and observations- derived fireline intensities, respectively.

to constrain and evaluate the approach (see Table B1). In this section, we step back and consider all performed simulations, to determine whether the same equations can also be used to classify penetrative vs. non-penetrative plumes.

The synthetic dataset described in 2.2 consisted of 140 runs and excluded 7 simulations, where the plume remained trapped in the ABL (see Table B1 and Table 3). We determined this by visual analysis of CWI centerline and smoke fields. The excluded plumes typically exhibited oscillatory or irregular centerline behavior (within the ABL, such as shown in the example in Fig. B1) with little or no smoke injected above $z_i$. For several combinations of fire and atmospheric conditions, however, making the distinction was challenging. For this reason, we included these "marginally-penetrative" plumes in the dataset.

In real-world applications, classification is a fundamental first step in plume rise parameterization process Sofiev et al. (2012). A viable automated method for categorizing penetrative vs. non-penetrative plumes requires that the distinction be made based on available input parameters, rather then smoke observations (as such are typically not available at the time of making a forecast).

Conveniently, we can use Eq. (8) to obtain a $z_{CL}$ estimate for any combination of input parameters without prior knowledge of plume type. It can, hence, be applied as a classifier by requiring that for a penetrative plume

$$z_{CL} > z_{i+} \tag{11}$$

where $z_{i+}$ denotes the height of the upper edge of the numerical grid box (or ambient atmospheric sounding) containing $z_i$. In other words, this definition ensures that $z_i$ and $z_{CL}$ are not in the same vertical model level. If this condition is not satisfied, the plume is assumed to be non-penetrative.

This approach correctly classifies all non-penetrative plumes that had been identified by visual analysis (Table 3). In addition, several plumes exhibiting marginal behavior are also classified by Eq. (11) to be non-penetrative.

For the purpose of subsequent dispersion modelling within real-world applications, non-penetrative plumes (i.e. all plumes listed in Table 3) would be assumed to become uniformly mixed in the vertical within a few convective turnover distanced downwind of the fire. Turbulent eddies within the ABL produce a well-mixed layer, resulting in relatively homogeneous vertical distribution of pollutants between the surface and $z_i$. In contrast, for plumes that extend above $z_i$, spanning the ABL, the entrainment layer, and/or the free troposphere, subsequent dispersion is typically handled by trajectory models.

## 5.2 Comparison with Existing Models

The above model evaluation indicates encouraging performance for the proposed smoke injection parameterization (Eq. (8)) at little computational cost. An additional advantage of our method is that it does not require making simplifying assumptions regarding the shape and heat flux distribution of the fire. This allows us to easily apply our model to complex heat sources, such as one produced with the strip head fire ignition pattern during the RxCADRE L2G prescribed burn (Fig. 8).

Unlike most existing plume rise parameterizations (Briggs, 1975; Rio et al., 2010; Freitas et al., 2007) we focus on a CWI centerline. Our model can be viewed as a "bulk method", having some common ground with the thermodynamic approach used in the FireWork modelling framework (Anderson et al., 2011; Chen et al., 2019) and the energy balance approach proposed by Sofiev et al. (2012). More specifically, we make no attempt to predict the full evolution of the rising plume centerline velocity or temperature before it reaches its equilibrium height. Rather, we focus on the energy balance of the plume over a "penetration layer".

**Table 3.** Identifying non-penetrative plumes using visual analysis vs. automated classification. Plume name denotes wind condition **W**, fuel type **F** and initial atmospheric profile **R**.

| Plume | Visual analysis | Automated classification |
|---|:---:|:---:|
| W5F9R1 | ✓ | ✓ |
| W5F1R3 | ✓ | ✓ |
| W5F8R3 | ✓ | ✓ |
| W5F9R3 | ✓ | ✓ |
| W5F1R7 | ✓ | ✓ |
| W5F8R7 | ✓ | ✓ |
| W5F9R7 | ✓ | ✓ |
| W5F1R0 | | ✓ |
| W5F1R1 | | ✓ |
| W5F8R1 | | ✓ |
| W5F10R3 | | ✓ |
| W5F11R3 | | ✓ |
| W5F1R4 | | ✓ |
| W5F11R4 | | ✓ |

Through analysis of the 140 LES experiments for plumes under variable fire and atmospheric conditions, we found that near-surface and boundary-layer plume dynamics are extraordinarily complex. While some aspects of plume mixing can be reasonably accounted for by making traditional entrainment assumptions, complicated features resulting from fire-atmosphere coupling, such as formation of lateral vortices and fireline wind convergence zone, are difficult to parameterize directly. Hence, we apply the energy balance approach to a layer well above the surface, starting from a reference height $z_s$ close to the top of the ABL.

As noted in Sect. 3, the implicit functional form of our solution (Eq. (8)) can be interpreted as a characteristic timescale multiplied by the characteristic velocity scale $w_f$. By rearranging Eq. (7) and substituting Eq. (8) for $z'$ it can be shown that the two expressions for $w_f$ are equivalent, namely:

$$w_f = \left[ \frac{I}{z_i \theta'} \right] = \left[ \frac{gIz'}{\theta_s z_i} \right]^{\frac{1}{3}} \tag{12}$$

The scaling relationship between vertical plume velocity and cubic root of fire heat has been previously established with both Rio's and Freita's models (Rio et al., 2010; Freitas et al., 2007), although our formulation includes different variables inside the radical. While both of our forms for $w_f$ and both model formulations (the simplified Eq. (7) and the expanded Eq. (8)) are mathematically equivalent, conversion from one form to another requires raising terms to 6[th] power. This results in large prediction errors; hence, for practical applications, the full Eq. (8) should be used.

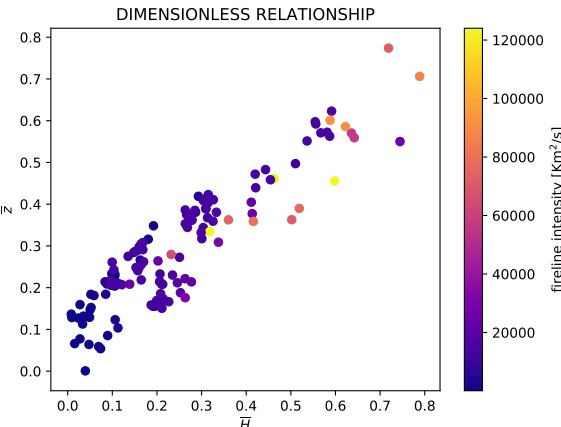

**Figure 10.** Similarity solution for dimensionless groups $\overline{H}$ and $\overline{z}$, corresponding to the RHS and LHS of Eq. (13), respectively. Scatter points represent individual LES runs, colored by fireline intensity parameter $I$.

### 5.3 Dimensionless Relationship

As discussed in Sect. 3, we can obtain an explicit solution for $z_{CL}$ by making additional assumptions about the vertical profile of potential temperature above the ABL. This allows us to reduce our Eq. (9) to a similarity relationship with two dimensionless

groups $\overline{z}$ and $\overline{H}$, denoting the left hand side (LHS) and right hand side (RHS) of Eq. (13), respectively. Nondimensional $\overline{z}$ and $\overline{H}$ are linearly related, as shown in Fig. 10. The simple relationship suggests that our modelling results could fairly easily be scaled to a wider range of fire and atmospheric conditions, beyond those captured by the synthetic dataset presented in the paper.

$$\underbrace{\frac{z'}{z_i}}_{\overline{z}} = \underbrace{C^{\frac{3}{2}} \left[ \frac{\theta_s}{g\gamma^3} \right]^{\frac{1}{4}} \left[ \frac{I}{z_i^3} \right]^{\frac{1}{2}}}_{\overline{H}} \tag{13}$$

### 5.4 Model Bias

The raw, non-bias- corrected form of the model suffers from a positive bias for tall plumes, as suggested by Fig. 5c and 6c. In other words, $z_{CL}$ is overpredicted for plumes injected high above the ABL. We speculate that this is due to the simplifying assumption that most of the cooling, mixing, and dilution occurs below the reference level $z_s$ in upper portion of the ABL.

As the distance between $z_s$ and $z_{CL}$ increases for tall plumes and as the smoke travels further into the free atmosphere,

this assumption becomes increasingly less accurate. Additional radiative cooling and entrainment of ambient air is, therefore, unaccounted for, resulting in over-prediction for $z_{CL}$.

This issue is largely resolved for our dataset with the applied bias-correction. However, cases with strong shear turbulence and active smoke mixing above the ABL are still likely to be overestimated.

## 5.5 Limitations

The most significant limitation of the proposed smoke injection height parameterization is that it applies only to smoke plumes with no water vapor condensation. Latent heat effects are not considered. Hence, smoke injection level for extreme pyroconvective events (e.g. flammagenitus clouds (WMO, 2017)) will likely be grossly under-predicted with the given formulation. Therefore, in its current form, our parameterization is unlikely to be suitable for large-scale applications (e.g. global chemical transport models). However, it has the potential to improve regional air quality tools (e.g. BlueSky), since wildfire emissions

sources are largely dominated by in- or near- ABL non-condensing smoke plumes (Val Martin et al., 2010, 2018).

Given the energy-balance formulation of our plume rise parameterization, it may be possible to incorporate latent heat effects by including an extra PE term in Eg. (1). Similarly to the iterative process for finding a level of neutral buoyancy with Eq. (8) using potential temperature, it may be possible to predict plume condensation level using ambient humidity profile. However, a big obstacle to this development is that, to our knowledge, WRF-SFIRE has not been validated for such conditions.

Unlike many existing methods, our parameterization relies on fireline intensity parameter $I$, rather than average fire heat flux values, as input. While this approach offers an advantage for modelling plumes from complex ignition sources (such as shown in Fig. 8), fireline intensity is difficult to observe in the field.

Another limitation is the inherently implicit form of the full model Eq. (8). While we have not encountered any issues using an iterative solver to find $z_{CL}$, atypical (or extremely noisy) ambient atmospheric soundings could potentially affect

convergence. The explicit form (Eq. (9)) derived using the idealizing ambient sounding (Fig. 4) offers a possible solution for such cases. However, it fails for weakly stable and adiabatic free atmosphere (eg. condition R8 in Fig. 1), as $\theta_s$ is extrapolated into lower levels of ABL.

Lastly, the model has been developed and tested only for typical daytime atmospheric conditions. We have not assessed model performance for stable night-time atmospheric profiles or in the presence of strong vertical windshear.

## 6 Conclusions

Plume rise estimation remains one a weak link in our ability to forecast where and how smoke from wildfires travels in the atmosphere. In this study we present a simple parameterization (Eq. (8)) for predicting CWI smoke-plume centerline height from a wildfire of an arbitrary shape and intensity. Our approach is based on energy-balance of the plume over a penetration region. We constrain and evaluate the proposed method using a synthetic LES-derived plume dataset developed for a wide

range of fire and atmospheric conditions.

Based on the results of cross-evaluation with LES data as well as a real prescribed burn case study, the parameterization offers reasonable accuracy at little computational cost. We demonstrate that the approach can also be applied as a classifier to distinguish penetrative and non-penetrative plumes. This information is key for subsequent dispersion modelling, as plume

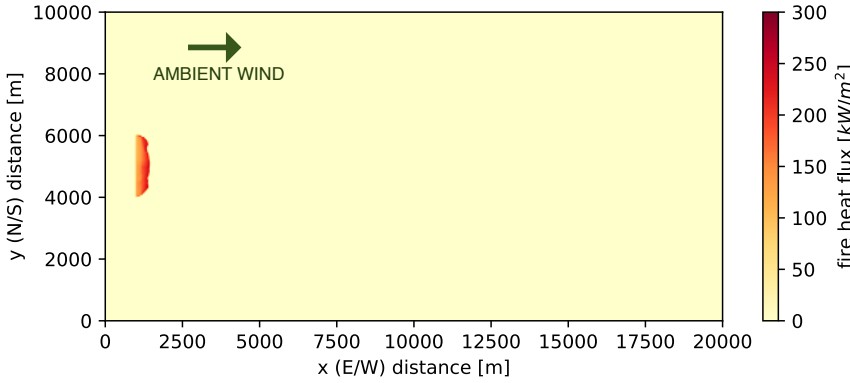

**Figure A1.** Numerical domain setup.

behavior is governed by different physics above and below the ABL. The proposed method can be used as a sand-alone
deterministic model or embedded in a host smoke modelling framework.

We hope that parameterization presented in this study will be of interest to air-quality researchers to provide a low-cost
solution for regional wildfire emissions-modelling applications.

*Code availability.*   S1: WRF-SFIRE sample initialization files (sample_simulation.zip)

## Appendix A:  Domain Setup

**Appendix B:  Parameter space of LES dataset**

Table B1 and Table B2 summarize the tested combinations of fire and atmospheric parameters captured by the synthetic plume
dataset. Colored cells correspond to completed simulations. Tall boundary layers of **R5** and **R6** domains required low winds
(5 ms$^{-1}$ and below) and high intensity fires (fuel categories 4, 6, 7, 12 and 13) to reach ABL top within the simulation
runtime and/or avoid smoke recirculation. Hence, alternative combinations (white cells in **R5** and **R6** columns) would require
considerably different domain setup from other runs. For this reason these combinations were not tested. Also, a single run was
performed for **R8** condition (adiabatic free atmosphere) as an extreme case scenario.

Red cells in Tab. B2 highlight simulations that were completed, but subsequently excluded from analysis presented in Sect.
3. This was done based on visual inspection of LES fields. There were two possible reasons for exclusion: (i) the plume reached
the top of the domain or (ii) the plume appeared to be non-penetrative. In the former case, it's questionable whether the fields
are physical, as the plume could potentially be affected by the absorbing layer near domain top, designed to prevent numerical
instability. The latter rendered the plume irrelevant for the purpose of analysis presented in Sect. 3. These non-penetrative runs,
however, were included for testing the plume classification method presented in Sect. 5.1.

**Table B1.** Combinations of test conditions resulting in penetrative plumes, as captured by the LES datasets. Green cell highlights fireline length condition (**L**) runs. Intensity of blue color corresponds to the number of runs for fuel condition (**F**) represented by the cell. Row 'W5' is expanded in Table B2 below.

| R/W | R0 | R1 | R2 | R3 | R4 | R5*† | R6*† | R7* | R8* |
|---|---|---|---|---|---|---|---|---|---|
| W3 | F7 | F7 | F7 | F7 | F7 | F7 | F7 | F7 | |
| W4 | F7 | F7 | F7 | F7 | F7L1 F7L2 F7L4 | F7 | F7 | F7 | |
| W5 | F1 - F12, excl:F4 | F1 - F13, excl:F9 | F1 - F13 | F2 - F13, excl:F8,F9 | F1 - F13 | F4 F6 F7 F12 F13 | F4 F6 F7 F12 F13 | F2 - F13, excl:F8,F9 | F7 |
| W6 | F7 | F7 | F7 | F7 | F7 | | | F7 | |
| W7 | F7 | F7 | F7 | F7 | F7 | | | F7 | |
| W8 | F7 | F7 | F7 | F7 | F7 | | | F7 | |
| W9 | F7 | F7 | F7 | F7 | F7 | | | F7 | |
| W10 | F7 | F7 | F7 | F7 | F7 | | | F7 | |
| W11 | F7 | F7 | F7 | F7 | F7 | | | F7 | |
| W12 | F7 | F7 | F7 | F7 | F7 | | | F7 | |

*Deep domain (5 km). †Extended runtime (30 min).

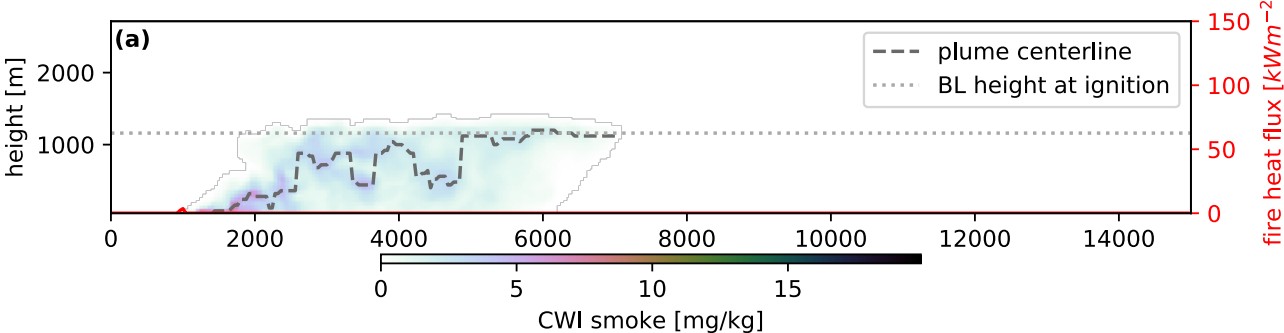

**Figure B1.** Fixed aspect ratio plot of CWI smoke from a sample non-penetrative plume (W5F8R3). Plume centerline and $z_i$ shown in dashed and dotted grey, respectively.

**Table B2.** Tested combinations of fuel and ABL conditions (all blue and red colored cells).

| R/W | R0 | R1 | R2 | R3 | R4 | R5*† | R6*† | R7* | R8* |
|---|---|---|---|---|---|---|---|---|---|
| F1 | | | | ABL plume | | | | ABL plume | |
| F2 | | | | | | | | | |
| F3 | | | | | | | | | |
| F4 | smoke at domain top | | | | | | | | |
| F5 | | | | | | | | | |
| F6 | | | | | | | | | |
| F7 | | | | | | | | | |
| F8 | | | | ABL plume | | | | ABL plume | |
| F9 | | ABL plume | | ABL plume | | | | ABL plume | |
| F10 | | | | | | | | | |
| F11 | | | | | | | | | |
| F12 | | | | | | | | | |
| F13 | smoke at domain top | | | | | | | | |

*Deep domain (5 km). †Extended runtime (30 min).

## Appendix C: Identifying Quasi-Stationarity

We define the quasi-stationary downwind region for each plume based on two factors: the height of the centerline and tracer concentration gradient along the centerline. Our filter attempts to extract only those portions of the downwind CWI smoke distribution, where both of these factors are changing slowly.

First, we remove the effect of random turbulent oscillations by applying a smoothing function (Savitzky-Golay filter provided by SciPy library with polynomial order set to 3) to both the concentration gradient along the centerline and the centerline height. We vary the size of the smoothing window as a function of mean ambient wind condition $\mathbf{W}$, such that $window\_length = max(\mathbf{W} \cdot 10 + 1, 51)$ grid points.

The filter then applies the following criteria to extract quasi-stationary regions:

– smoothed tracer concentration along the plume centerline varies by less then 10% of the maximum concentration gradient

– smoothed centerline height varies by less then a 100 m

– the location is downwind of the maximum tracer concentration gradient

**Table D1.** Variable descriptions and units used in smoke injection model.

| Variable | Unit | Description |
|----------|------|-------------|
| $I$ | $\mathrm{Km^2s^{-1}}$ | fireline integrated heat flux |
| g | $\mathrm{ms^{-2}}$ | gravity constant = 9.81 |
| $\theta_{CL}$ | K | ambient potential temperature at $z_{CL}$ |
| $\theta_s$ | K | ambient potential temperature at $z_s$ |
| $z_{CL}$ | m | smoke injection height |
| $z_i$ | m | boundary layer height |
| $z_s$ | m | reference height |

– the location is at least 10 grid points away from the maximum in smoothed and non-smoothed centerline height

    – the location is at least 50 grid points away from the downwind endpoint of the centerline

The above thresholds were determined through an informal sensitivity analysis (not shown), based on the filter's ability to effectively identify regions of near-stationary plume centerline height for all simulations in our dataset.

## Appendix D: Estimating Model Input Parameters

Summarized in Table D1 are parameters associated with an iterative solution for $z_{CL}$ using Eq. (8). Below is our approach to estimating these parameters from LES data.

As noted above, we consider the problem in crosswind direction. Given a three-dimensional fire of an arbitrary shape (eg. Fig. 2b) and an ambient atmospheric sounding, we first average the fire kinematic heat flux for all ignited cells (where heat flux > 1 kWm$^{-2}$) over the crosswind (y) direction at the surface (red line on Fig. 2a). Due to surface wind shear this direction

may differ from the one used for calculating CWI smoke concentrations (as shown in Sect. 4.3). To obtain fireline intensity parameter $I$ we numerically integrate the crosswind averaged heat fluxes over the depth of the fireline in the along-wind (x) direction.

We use pre-ignition potential temperature profile (i.e. the ambient environment upwind of the fire) averaged over the entire LES domain as an environmental sounding. All model fields are interpolated to have a 20 m vertical increment. $z_i$ is defined as

the height of the strongest environmental lapse rate gradient, and $z_s = \frac{3}{4}z_i$, based on informal model sensitivity analysis (not shown). The exact choice of $z_s$ has little effect on model performance as long as it remains within the upper portion of the uniform potential temperature well-mixed layer.

The values of $\theta_s$ and $\theta_{CL}$ are then determined from the pre-ignition sounding for each simulation using the definitions of $z_s$ and $z_{CL}$ (as described in Sect. 2.3).

## Appendix E: Iterative Solution for $z_{CL}$

The numerical implementation of our iterative solution using SciPy's fsolve function (scipy.optimize.fsolve) is as follows. We rewrite bias corrected Eq. (8) into an input function $toSolve$ as:

$$toSolve = lambda\ z : z - B_1(z_s + C\left[\frac{g(T0[int(\frac{z}{dz})] - \theta_s)}{\theta_s(z - z_s)}\right]^{-\frac{1}{2}}\left[\frac{gI(z - z_s)}{\theta_s z_i}\right]^{\frac{1}{3}}) - B_2 \tag{E1}$$

where $C = 1.005$, $B_1 = 0.924$ and $B_2 = 116.417$ are bias correction parameters, $T0$ is the potential temperature sounding vector, $dz$ is the vertical step and $int()$ is a standard Python function converting the bracketed value into an integer.

A possible issue for some solvers is that we are, effectively, iterating over the vertical index of the column vector $T0$ corresponding to $z_{CL}$. As the numerical solver attempts to converge on a solution it may query a non-existent index and fail. We are able to obtain a fast and consistent performance by ensuring we set $z_i$ as the initial guess for $z_{CL}$ and by minimizing the initial step bound option of the solver

$$z_{CL} = fsolve(toSolve, z_i, factor = 0.1) \tag{E2}$$

*Author contributions.* Conceptualization: Nadya Moisseeva and Roland Stull; methodology: Nadya Moisseeva; resources: Roland Stull; data curation: Nadya Moisseeva; writing (original draft preparation): Nadya Moisseeva and Roland Stull; writing (review and editing): Nadya Moisseeva and Roland Stull; visualization: Nadya Moisseeva; supervision: Roland Stull; funding acquisition: Nadya Moisseeva and Roland Stull

*Competing interests.* The authors declare that they have no conflict of interest.

*Acknowledgements.* We sincerely thank Dr. Rosie Howard and Chis Rodell for the countless fruitful discussions, new ideas and encouragement. We would like to acknowledge WestGrid and ComputeCanada for providing computational resources for LES runs, and Julia Jeworrek for her ongoing generous help with cluster access. Thank you to all members of UBC Weather Research and Forecasting Team for their motivation and support. This work was funded by grants from Natural Sciences and Engineering Research Council of Canada (NSERC), Natural Resources Canada (NRCan), Fraser Basin Council (BC CLEAR), British Columbia Ministry of Environment and Climate Change Strategy, Alberta Ministry of Environment and Parks and Government of the Northwest Territories.

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
