# Peer review of "Wildfire smoke-plume rise: a simple energy balance parameterization"

_Atmospheric Chemistry and Physics, 2020_

## Referee Comment (RC1) · Anonymous Referee #1 · 22 Sep 2020

General remarks:

The authors present a new plume rise parameterization based on LES simulations using a synthetic plume-rise data set from WRF-SFIRE LES runs. The parameterization presented here is inexpensive and is able replicate fire plume rises that penetrate through the daytime PBL. Considering the current ongoing fires during the 2020 fire season, significant work is needed to better predict wildfire plume rises and smoke dispersion. As a result, the work carried out here is very timely and important.

With that said, the reviewer has some concerns in regard to the study presented here.

There are a number of plume rise modeling frameworks out there (Briggs, 1975; Sofeiv et al. 2012, Freitas et al. 2007; 2010), why develop yet another plume rise model? This

should be made clear in the introduction section. For example, what is meant by an idealized heat source? Does this statement refer to the plume geometry?

The biggest limitation of this study is that it does not consider moisture sources, especially in the ambient atmosphere. Studies have shown that PyroCb development is strongly dependent on mid-level moisture. While PyroCbs are somewhat rare, these plumes are often responsible for some of the largest mass injections (2018) of smoke in the atmosphere, with some of these rivaling that of significant volcanic eruptions. PyroCus, which are far more common than PyroCb, are also strongly impacted by the vertical moisture profile (albeit less so then PyroCbs).

It was noted by the authors that weakly buoyant plumes that do not penetrate into the free troposphere were not considered as part of the data set. Why did the authors use the RxCADRE L2G as a case study? Overall, most of the RxCADRE prescribe burns were lower intensity fires. The largest burn (L2F), had a plume rise height that only reached an altitude of 1.5 km, which was still below the PBL. The reviewer does understand that there are a limited number of data sets where the plume rise is measured with constrained lower boundary conditions (i.e heat fluxes)

The reviewer has concerns that this study may be too limited in scope. The authors do not include plumes that fall below the PBL in their data set while this parameterization is only applicable for plumes that do not reach the lifting condensation level, which is height where plumes would be driven latent heat releases. As a result, it must be concluded that this parameterization will only valid for plumes greater than the PBL but less than the LCL, which seems like a narrow range of plumes that this parameterization is appropriate for. Furthermore, PyroCu and PyroCb events are usually associated with large fires that emit a lot of pollutants at high altitudes. Usually, its these smoke emissions that last the longest in the atmosphere (Peterson et al. 2018; Christian et al. 2019) and are the pollutants that fire researchers and forecasters are probably most interested in.

As a result, the reviewer implores that the authors consider the effects of vertical moisture profiles within their parameterization. This work could be significantly more impactful if this limitation can be remedied as the parameterization presented here would have a clear advantage over the semi-empirical plume rise formulas discussed in Sofiev et al. (2012) and Briggs (1975).

Specific comments:

Line 49: Mallia et al. 2018 did not use WRF-SFIRE, so this may not be an appropriate citation for this particular statement. Mallia et al. 2020 did use WRF-SFIRE though in their analysis though (see citation below).

Lines 52-58: While WRF-SFIRE is a well-documented coupled fire-atmosphere model, the authors should consider expanding the description for this modeling framework. Not all reviewers may be familiar with WRF-SFIRE.

Line 56: Why were these key parameters selected? For example, why not include ambient moisture? A number of well-known studies have shown that ambient moisture profile can significantly impact fire plume rise development, especially for PyroCb development (Freitas et al. 2007; Peterson et al. 2017; Tory et al. 2018)

Lines 53-78: A figure showing the numerical setup could be helpful to add here.

Line 93-94: Can the authors provide recommendations on how atmospheric transport models should deal with weakly buoyant non-penetrative plumes that do not penetrate into the free troposphere?

Lines 200-206: Why use RxCADRE when the authors were excluding plume rises that fell below the PBL height? Most of the RxCADRE burns were relatively small with plume rises that only reached an altitude of 1300 m.

Lines 218-219: What time was the sounding relative to the start of the burn?

Lines 263: Is there a way that the author could test this hypothesis?

Line 281: This conclusion section is really limited. Perhaps the authors could better synthesize the results in their study?

In addition, this parameterization seems to be limited to plumes that fall above the PBL but less than the LCL (i.e cases where the is no PyroCu or PyroCb development). As a result, who is this parameterization geared towards? Why not just use the parameterization discussed in Freitas et al. 2007, which includes entrainment, wind shear (without restriction), and moisture effects. While the reviewer appreciates that this model can be run at a low computation cost, it seems like this parameterization comes with a number of cavaets that could limit its usefulness.

Figure 1: This is not referenced in the text, except in the conclusion section.

Figure 2a: Might be worthwhile to add distance to the x-axis here even though it is done in

Figure 2b. Took a second to figure out what the x-axis was showing. The reviewer also recommends switching panels b and d since panel b corresponds to the plume in cross section (panel a)

References:

Christian, K., Wang, J., Ge, C., Peterson, D., Hyer, E., Yorks, J., & McGill, M. (2019). Radiative forcing and stratospheric warming of pyrocumulonimbus smoke aerosols: First modeling results with multisensor (EPIC, CALIPSO, and CATS) views from space. Geophysical Research Letters, 46, 10,061–10,071. https://doi.org/10.1029/2019GL082360

Mallia, D.V.; Kochanski, A.K.; Urbanski, S.P.; Lin, J.C. Optimizing Smoke and Plume Rise Modeling Approaches at Local Scales. Atmosphere 2018, 9, 166.

Peterson et al. (2017): A Conceptual Model for Development of Intense Pyrocumulonimbus in Western North America. DOI: 10.1175/MWR-D-16-0232.1

[Figure]

Peterson et al. (2018): Wildfire-driven thunderstorms cause a volcano-like strato-spheric injection of smoke

Tory et al. (2018) Thermodynamics of Pyrocumulus: A Conceptual Study. MWR.

---

## Author Comment (AC1) · 23 Oct 2020

We sincerely thank the Reviewer for their comments and insights. Please find our responses structured as follows:
- Original Reviewer comments in ***bold italics***
- Author responses as regular text
- Manuscript edits and changes in blue

**Reviewer #1 Comments:**

***General remarks:***

***The authors present a new plume rise parameterization based on LES simulations using a synthetic plume-rise data set from WRF-SFIRE LES runs. The parameterization presented here is inexpensive and is able replicate fire plume rises that penetrate through the daytime PBL. Considering the current ongoing fires during the 2020 fire season, significant work is needed to better predict wildfire plume rises and smoke dispersion. As a result, the work carried out here is very timely and important.***

***With that said, the reviewer has some concerns in regard to the study presented here.***

***There are a number of plume rise modeling frameworks out there (Briggs, 1975; Sofeiv et al. 2012, Freitas et al. 2007; 2010), why develop yet another plume rise model? This should be made clear in the introduction section. For example, what is meant by an idealized heat source? Does this statement refer to the plume geometry?***

The main goal of developing another plume-rise parameterization is that existing approaches remain subject to large uncertainties. For example, evaluation of BlueSky performance suggests that correlation coefficients associated with smoke injection predictions (based on adapted Brigg's algorithm) are on the order of 0.1 (Raffuse et al. 2012). Similarly low $R^2$-values (maximum of 0.33) were noted for Freitas model (Val Martin et al., 2012). Moreover, existing tools struggle to reliably differentiate which plumes remain in the ABL and which penetrate it (Val Martin 2012): information, which is key for subsequent dispersion modelling (Sofief et al. 2012). We added the following clarification at the end of the Introduction section:

…We then demonstrate with both numerical and prescribed burn data, that within the range of tested conditions this parameterization offers high speed and accuracy (Sect. 4). Moreover, it provides the means for classifying penetrative vs. non-penetrative plumes, which is key for subsequent dispersion modelling (Sofiev et al., 2012, Val Martin et al., 2012).
    The proposed approach is geared toward regional smoke modelling frameworks (e.g. BlueSky and BlueSky Canada). Government agencies, air quality managers and fire response teams depend on these operational tools and their accuracy to issue air quality warnings, evacuation orders and to help mitigate human health impacts. Yet, model evaluation studies suggest that plume rise estimation remains a weak link within these smoke modelling systems (Raffuse et al., 2012, Val Martin et al. 2012, Chen et al. 2019). Moreover, existing methods struggle to reliably differentiate which plumes remain in the ABL and which penetrate. The broad goal of the work is, therefore, to address some of these challenges and improve the accuracy of plume rise predictions for regional air quality applications.

We added the following clarification regarding idealization of heat sources in Freita's and Rio's models:

…Notably, both consider an idealized heat source to represent the fire. To initialize the plume at the lower boundary, simplified fire geometry (circular and rectangular for Freita's and Rio's models, respectively) with a uniform heat flux is assumed.

*The biggest limitation of this study is that it does not consider moisture sources, especially in the ambient atmosphere. Studies have shown that PyroCb development is strongly dependent on mid-level moisture. While PyroCbs are somewhat rare, these plumes are often responsible for some of the largest mass injections (2018) of smoke in the atmosphere, with some of these rivaling that of significant volcanic eruptions. PyroCus, which are far more common than PyroCb, are also strongly impacted by the vertical moisture profile (albeit less so then PyroCbs).*

There are two important considerations regarding inclusion of moisture in the model (i) effects of ambient atmospheric moisture profile on plume development without saturation (ii) effects of latent heat release in plumes reaching condensation. Our proposed parameterization makes no assumptions regarding the former (e.g. the ambient sounding of the observational case study included moisture). The latter indeed remains to be addressed.

We certainly agree with the Reviewer that no-condensation assumption is the most significant limitation of the proposed model (as we explicitly state on Lines 270-274 of the original manuscript version). Apart from its accuracy, the reason we feel our parameterization is useful even in its current form is that vast majority of plumes are non-condensing. Based on remotely-sensed estimates, only 4-12% of the plumes over North America reach PBL top; amongst those that do, most (>83%) remain in the stable layers directly above PBL top (Val Martin 2010). While we are not aware of a formal climatology of pyroconvective events, analysis of a sample of remotely-sensed *high-altitude* plumes (above 5500m) over Yukon and Alaska suggests that less than 3% resulted in PyroCu formation (Val Martin 2010). This renders such events extremely rare.

Moreover, while formation of pyroconvective clouds undoubtedly effects *maximum* plume rise, it's effect on mean smoke injection height is less clear. Figure 1 below shows a plume height retrieval from MISR satellite (using MINX software) associated with Silver Fire in New Mexico in July 2013 (Nelson 2013). Notably, mean smoke injection height (subplot (c)) is largely unaffected by the presence of PyroCu.

[Figure]

**Figure 1. MINX plume height retrieval software output associated with the Silver Fire in New Mexico on MISR orbit 71726, block 63–64, from 12 July 2013 (Nelson 2013) (a) MISR nadir RGB image (b) MINX height retrievals (c) MINX height retrieval profiles**

While inclusion of condensation is important for global chemical transport modelling application, regional smoke modelling frameworks (towards which this parameterization is geared) often do not consider moisture effects (e.g. BlueSky, BlueSky Canada, FireWork). That being said, we recognize the importance of extreme pyroconvective events, in particular for large scale applications. Detailed reasons for their exclusion from our current formulation as well as manuscript additions are addressed in comments below.

*It was noted by the authors that weakly buoyant plumes that do not penetrate into the free troposphere were not considered as part of the data set. Why did the authors use the RxCADRE L2G as a case study? Overall, most of the RxCADRE prescribe burns were lower intensity fires. The largest burn (L2F), had a plume rise height that only reached an altitude of 1.5 km, which was still below the PBL. The reviewer does understand that there are a limited number of data sets where the plume rise is measured with constrained lower boundary conditions (i.e heat fluxes)*

Apart from the uniquely comprehensive nature of the dataset, the reason we consider RxCADRE L2G is because it was indeed a penetrative plume (see response to Reviewer's "Specific Comments").

*The reviewer has concerns that this study may be too limited in scope. The authors do not include plumes that fall below the PBL in their data set while this parameterization is only applicable for plumes that do not reach the lifting condensation level, which is height where plumes would be driven latent heat releases. As a result, it must be concluded that this parameterization will only valid for plumes greater than the PBL but less than the LCL, which seems like a narrow range of plumes that this parameterization is appropriate for. Furthermore, PyroCu and PyroCb events are usually associated with large fires that emit a lot of pollutants at high altitudes. Usually, its these smoke emissions that last the longest in the atmosphere (Peterson et al. 2018; Christian et al. 2019) and are the pollutants that fire researchers and forecasters are probably most interested in.*

The presented parameterization is indeed valid for most (including PBL) plumes. We thank the Reviewer for pointing out that this was not made clear in the manuscript and hope that our additions address Reviewer's concerns.

While we did simulate boundary layer plumes (e.g. Figures 2 and 3 below), we excluded them from our training dataset because the concept of "injection height" is not relevant for non-penetrative cases. The smoke from such plumes becomes uniformly mixed throughout the depth of the turbulent PBL within several convective turnover periods. Often these low-buoyancy plumes initially exhibit irregular or oscillatory centerline behavior, rendering "injection height" concept irrelevant (see Figures 2 and 3 below). However, knowing which plumes will penetrate the PBL and which ones will not is critical for dispersion modelling (Val Martin et al. 2018, Sofiev et al. 2012). Our parameterization allows to determine this with encouraging accuracy. We, hence, revised our manuscript to include Section 5.1 and Appendix B, dedicated to LES simulations of non-penetrative and plume classification using our parameterization.

[Figure]

**Figure 2. Oscillatory behavior of a sample non-penetrative plume in the boundary layer.**

[Figure]

**Figure 3. Irregular centerline behavior of a sample non-penetrative plume in the boundary layer.**

Regarding condensation effects (PyroCu and PyroCb), we certainly acknowledge this as a limitation (see response to comment above). However, while our parameterized injection heights for such extreme events are likely to be underestimated, given typical prediction errors associated with existing models (on the order of kilometers) (Raffuse et al. 2012, Val Martin et al. 2012), they may well provide comparable performance.

Manuscript changes: Addition of Section 5.1 and Appendix B

*As a result, the reviewer implores that the authors consider the effects of vertical moisture profiles within their parameterization. This work could be significantly more impactful if this limitation can be remedied as the parameterization presented here would have a clear advantage over the semi-empirical plume rise formulas discussed in Sofiev et al. (2012) and Briggs (1975).*

As noted in our earlier response, our parameterization does not make assumptions about the vertical moisture profile for non-condensing plumes. We have explicitly considered ambient moisture in early model development stage. Figure 4 shows plume profiles under 'dry' and 'wet' (approximately 80% relative humidity, depending on the vertical level) environmental conditions, with the remaining parameters held constant. Note that the plumes are nearly identical, subject to random turbulence. The prediction error (obtained using iterative solution of our model) was 64 m for the 'dry' case and 14 m for the 'wet' case. Ambient moisture was also included in the evaluation case study (Section 4.3).

[Figure]

**Figure 4. Plumes under 'dry' and 'wet' ambient atmospheric conditions (with remaining conditions held constant).**

While we have confirmed that ambient moisture (without condensation) has little impact on our model performance, we agree with the Reviewer regarding the possible effects of PyCu. However, in order to include condensation and cloud formation, we first have to determine whether the LES can reasonably capture such conditions. To our knowledge, no such validation studies are available yet. Moreover, this modelling effort would require an evaluation dataset constraining not only the heat source and cloud development, but also the moisture fluxes parameterized by SFIRE. We hope that the current efforts of the FASMEE campaign (Prichard et al 2019) could make this possible in the near future. We added the following discussion to the Limitations section of the manuscript:

…The most significant limitation of the proposed smoke injection height parameterization is that it applies only to smoke plumes with no water vapor condensation. Latent heat effects are not considered. Hence, smoke injection level for extreme pyroconvective events (e.g. flammagenitus clouds (WMO 2017) will likely be grossly under-predicted with the given formulation. Therefore, in its current form, our parameterization is unlikely to be suitable for large-scale applications (e.g. global chemical transport models). However, it has the potential to improve regional air quality tools (e.g. BlueSky), since wildfire emissions sources are largely dominated by in- or near- ABL non-condensing smoke plumes (Val Martin et al., 2010, 2018).
        Given the energy-balance formulation of our plume rise parameterization, it may be possible to incorporate latent heat effects by including an extra PE term in Eg. (1). Similarly to the iterative process for finding a level of neutral buoyancy with Eq. (8) using potential temperature, it may be possible to predict plume condensation level using ambient humidity profile. However, a big obstacle to this development is that, to our knowledge, WRF-SFIRE has not been validated for such conditions.

***Specific comments:***

***Line 49: Mallia et al. 2018 did not use WRF-SFIRE, so this may not be an appropriate citation for this particular statement. Mallia et al. 2020 did use WRF-SFIRE though in their analysis though (see citation below).***

Thank you for highlighting the error. We've changed the citation to Mallia et al. 2020.

***Lines 52-58: While WRF-SFIRE is a well-documented coupled fire-atmosphere model, the authors should consider expanding the description for this modeling framework. Not all reviewers may be familiar with WRF-SFIRE.***

We added the following short description to the manuscript at the beginning of Section 2:

…The model allows to explicitly resolve plume dynamics, while parameterizing fuel combustion. One of the primary advantages of using WRF-SFIRE is that it supports two-way coupling between the atmosphere and the fire behavior model, allowing it to capture some of the complex dynamical feedbacks that exist between the fire and the atmosphere (Prichard et al. 2019). Heat and moisture fluxes from the simulated burn provide forcing to the atmosphere, affecting local wind flow and thermodynamics. This in turn influences the modelled fire behavior.

***Line 56: Why were these key parameters selected? For example, why not include ambient moisture? A number of well-known studies have shown that ambient moisture profile can significantly impact fire plume rise development, especially for PyroCb development (Freitas et al. 2007; Peterson et al. 2017; Tory et al. 2018)***

Key parameter choices were largely dictated by what would be available as input from the host air quality model, as well as what has been broadly recognized as relevant in the literature. Our early simulations during model development also included various ambient surface heat fluxes (to examine the effect of ABL mixing) and moisture (see above). However, both of these factors did not appear to have a direct effect on our parameterization. We added the following clarification in the manuscript:

…Table 2 summarizes the key parameters that were varied to produce the synthetic dataset. The reason for considering the given conditions is twofold: these parameters (i) have been widely acknowledged as having a strong impact on plume behavior and (ii) can be obtained (and provided as input for the parameterization) under real-world scenarios.

Other manuscript changes (regarding condensation): Expanded Limitations section as per responses above.

**Lines 53-78: A figure showing the numerical setup could be helpful to add here.**

We added an illustration of the domain setup (and a corresponding in-text reference) as Appendix A.

**Line 93-94: Can the authors provide recommendations on how atmospheric transport models should deal with weakly buoyant non-penetrative plumes that do not penetrate into the free troposphere?**

Please see the response to a related comment above.

Manuscript changes: Addition of Section 5.1 on plume classification.

**Lines 200-206: Why use RxCADRE when the authors were excluding plume rises that fell below the PBL height? Most of the RxCADRE burns were relatively small with plume rises that only reached an altitude of 1300 m.**

As note in an earlier response, L2G burn indeed produced a penetrative plume. While the Reviewer is correct to note that the plume's maximum altitude may not be impressive, the boundary layer top during the burn was at roughly 1060 m (AGL). We added the following clarification in Section 4.3 of the manuscript:

…We use observational data from the RxCADRE L2G prescribed burn (Ottmar et al., 2016) and it's numerical simulation (Moisseeva and Stull, 2019). This case study was selected based on (i) comprehensive nature of the observational dataset (ii) penetration of the plume above ABL top and (iii) availability of a completed model validation study, confirming that WRF-SFIRE reasonably captures the smoke plume produced during the burn.

**Lines 218-219: What time was the sounding relative to the start of the burn?**

The pre-ignition sounding for L2G was performed at 10:00CST (i.e. 2h23min prior to the burn). We've added this in the manuscript:

…Due to wind shear, as measured by the sounding launched prior to the burn (10:00:00 CST), the CWI direction at the surface differs from the one used to estimate CWI smoke.

**Lines 263: Is there a way that the author could test this hypothesis?**

One way to examine whether additional mixing occurs above $z_s$ is to consider conserved variable plots for plume centerlines. Figure 5 below compares plumes with shallow and deep penetration depths (note, the scales differ for both axes).

[Figure]

**Figure 5. Conserved variable plots for shallow- and deep- penetrating plumes. Scatter point color correspond to normalized (by $z_i$) height of the centerline at the given location.**

For the shallow case (left), little additional cooling occurs beyond boundary layer top (when the scatter points turn grey at $z/z_i=1$). For the deep case (right), there is a much more noticeable overshoot (deep red) of the centerline equilibrium height and obvious additional cooling occurring before $z_{CL}$ is reached (light red). This generally supports our hypothesis. However, we do not see this occurring for all plumes. In part, this is expected, as the magnitude of the bias is on the order of general scatter in the data (see Figure 3 in the original manuscript). Hence, we do not feel we have sufficient evidence to draw conclusions.

***Line 281: This conclusion section is really limited. Perhaps the authors could better synthesize the results in their study?***

We edited the Conclusions section as follows:

Plume rise estimation remains one of the weakest links in our ability to forecast where and how smoke from wildfires travels in the atmosphere. In this study we present a simple parameterization (Eq. (8)) for predicting CWI smoke-plume centerline height from a wildfire of an arbitrary shape and intensity. Our approach is based on simple energy-balance of the plume over the penetration region. We constrain and evaluate the proposed method using a synthetic LES-derived plume dataset developed for a wide range of fire and atmospheric conditions.

Based on the results of cross-evaluation with LES data as well as a real prescribed burn case study, the parameterization offers reasonable accuracy at little computational cost. We demonstrate that the approach can also be applied as a classifier to distinguish penetrative and non-penetrative plumes. This information is key for subsequent dispersion modelling, as plume behavior is governed by different physics above and below the ABL. The proposed method can be used as a sand-alone deterministic model or embedded in a host smoke modelling framework.

We hope that parameterization presented in this study will be of interest to air-quality researchers to provide a low-cost solution for regional wildfire emissions-modelling applications.

***In addition, this parameterization seems to be limited to plumes that fall above the PBL but less than the LCL (i.e cases where the is no PyroCu or PyroCb development). As a result, who is this parameterization geared towards? Why not just use the parameterization discussed in Freitas et al. 2007, which includes entrainment, wind shear (without restriction), and moisture effects. While the reviewer appreciates that this model can be run at a low computation cost, it seems like this parameterization comes with a number of cavaets that could limit its usefulness.***

We hope our previous comments addressed model applications.

This parameterization is geared towards regional air quality systems. Namely, it is most appropriate for smoke modelling frameworks, such as BlueSky and BlueSky Canada (which currently rely on adapted Briggs algorithms). We certainly agree that Freitas model is much better suited for global chemical transport modelling applications then our method.

Manuscript changes: Expanded Introduction section (see response to Reviewer's first general comment).

The overall reason for not using existing parameterisations is their somewhat discouraging performance (see response to Reviewer's first general comment). This can partly be partly attributed to uncertainties in fire input parameters - a source of error we don't have to worry about in LES simulations. However, parameterization of entrainment is actually a substantial limitation within prognostic models (including assumed instantaneous mixing along an idealized radially symmetric plume, constant empirical entrainment parameters, assumption of proportionality between vertical plume velocity and entrainment rate, among many others). These entrainment assumptions date back to tank experiments of Morton and Turner (Morton et al. 1956) for point buoyancy sources in uniformly stratified fluids, and it's questionable how applicable they are for wildfires. In part, explicit parameterization of windshear introduced in the later version of Freitas model aimed to remedy some of the limitations of the original entrainment assumptions (Freitas 2010). This is where LES can again offer a significant advantage, hence, its common use in other entrainment-focused fields, such as cloud physics (Dawe and Austin 2013).

***Figure 1: This is not referenced in the text, except in the conclusion section.***

The figure is referenced on line 90 (Section 2.2) of the original manuscript.

***Figure 2a: Might be worthwhile to add distance to the x-axis here even though it is done in Figure 2b. Took a second to figure out what the x-axis was showing. The reviewer also recommends switching panels b and d since panel b corresponds to the plume in cross section (panel a)***

We added the axes labels, as requested.

The main reason for the chosen subplot order is that the cross-section shown in panel (d) is based on the shaded quasi-stationary region identified in sublot (c). Therefore, we feel it's necessary to keep the order as is. We added a clarification in the caption:

…(d) Representative downwind smoke distribution. The profile (solid blue line) is obtained by horizontally averaging the CWI smoke concentrations in the quasi-stationary region (dashed grey in (c)).

**References:**

Dawe, J. T., and P. H. Austin. "Direct entrainment and detrainment rate distributions of individual shallow cumulus clouds in an LES." *Atmospheric Chemistry & Physics* 13.15 (2013).

Freitas, S. R., et al. "Sensitivity of 1-D smoke plume rise models to the inclusion of environmental wind drag." *Atmospheric Chemistry & Physics* 10.2 (2010).

Morton, B. R., Geoffrey Ingram Taylor, and John Stewart Turner. "Turbulent gravitational convection from maintained and instantaneous sources." *Proceedings of the Royal Society of London. Series A. Mathematical and Physical Sciences* 234.1196 (1956): 1-23.

Nelson, David L., et al. "Stereoscopic height and wind retrievals for aerosol plumes with the MISR INteractive eXplorer (MINX)." *Remote Sensing* 5.9 (2013): 4593-4628.

Raffuse, Sean M., et al. "An evaluation of modeled plume injection height with satellite-derived observed plume height." *Atmosphere* 3.1 (2012): 103-123.

Sofiev, M., T. Ermakova, and R. Vankevich. "Evaluation of the smoke-injection height from wild-land fires using remote-sensing data." *Atmospheric Chemistry & Physics* 12.4 (2012).

Val Martin, Maria, Ralph A. Kahn, and Mika G. Tosca. "A global analysis of wildfire smoke injection heights derived from space-based multi-angle imaging." *Remote Sensing* 10.10 (2018): 1609.

Val Martin, M., Logan, J. A., Kahn, R. A., Leung, F. Y., Nelson, D. L., & Diner, D. J. "Smoke injection heights from fires in North America: analysis of 5 years of satellite observations". *Atmos. Chem. Phys*, *10*.4 (2010): 1491-1510.

---

## Referee Comment (RC2) · Anonymous Referee #2 · 9 Nov 2020

This paper introduces updraft velocity scales that are used during daytime CBLs. The scaling was developed using a set of model-derived synthetic plumes from WRF-SFIRE. The paper is novel and well-written. This reviewer feels that the observational dataset used is not ideal compared to other wildfire plume observations available. A limitation to the study and proposed methodology is the use of fireline intensity (heat flux) as this parameter is very difficult to observe in the field and even more so for wildfires. The data used are limited to the flame zone, but not the plume base. As the authors do recognize, the data from multiple sensors, have a range of values. It may be worthwhile to use sensible heat flux values calculated from the in situ tower observations. Overall, this is an excellent paper, well written and justified. I recommend publication after Minor Revisions.

While the proposed method for estimating plume rise is somewhat novel, it is unclear why the authors don't use more observational data. The authors state that observations are limited and that is somewhat true, but given the recent publication of key wildfire plume datasets (RaDFIRE; Clements et al. 2018), the authors should really use wildfire observations verses low-intensity prescribed fires from RxCADRE. Another issue with the methodology presented in this study is that the authors use a vertical velocity scale for plume rise, but have no vertical velocity observations. Vertical velocity data are also available from the RxCADRE dataset. Additionally, a very recent paper by Rodriguez et al. (2020) show deep updraft velocities in a megafire that could be used as an extreme boundary for the parameterization. Additionally, a dataset from Lareau and Clements (2017) of a wildfire that includes plume evolution in a cross-wind is available as was also used by (Mallia et al. 2019).

Some specific comments:

Line 148: It is not clear what the authors are defining as Fireline Intensity: ". . . fireline intensity parameter I, which is the the kinematic heat flux into the atmosphere integratedacross the fireline depth (in units of Km2s−1),. . ."

I would call this the fire heat flux vs Byram's Fireline Intensity which has units of kW /m.

Line 205: Replace "lot" with "plot."

In Figure 2a, the mean plume centerline has a loop just downwind of the initial injection. Is this realistic? I would imagine that this feature represents the CBL, but would be averaged out as observed in the remainder of the downwind plume. Can the authors comment on this structure and whether this is realistic?

---

## Author Comment (AC2) · 14 Nov 2020

We sincerely thank the Reviewer for their comments and insights. Please find our responses structured as follows:
- Original Reviewer comments in **_bold italics_**
- Author responses as regular text
- Manuscript edits and changes in blue

**Reviewer #2 Comments:**

**_General remarks:_**

**_This paper introduces updraft velocity scales that are used during daytime CBLs. The scaling was developed using a set of model-derived synthetic plumes from WRF-SFIRE. The paper is novel and well-written. This reviewer feels that the observational dataset used is not ideal compared to other wildfire plume observations available. A limitation to the study and proposed methodology is the use of fireline intensity (heat flux) as this parameter is very difficult to observe in the field and even more so for wildfires. The data used are limited to the flame zone, but not the plume base. As the authors do recognize, the data from multiple sensors, have a range of values. It may be worthwhile to use sensible heat flux values calculated from the in situ tower observations. Overall, this is an excellent paper, well written and justified. I recommend publication after Minor Revisions._**

We have explored using fire area and average heat flux instead of fireline intensity as input parameters for our model, however, we were not successful. We thank the Reviewer for pointing out this limitation. We added the following clarification in Section 5.5 (Limitations) of the manuscript:

Unlike many existing methods, our parameterization relies on fireline intensity parameter *I*, rather than average fire heat flux value, as input. While this approach offers an advantage for modelling plumes from complex ignition sources (such as shown in Fig. 8), fireline intensity is difficult to observe in the field.

We address observational datasets in the next comment.

While we are aware of the anemometer data from L2G in-situ tower, we find it's challenging to use as a means of quantifying fire behavior. Estimated sensible heat flux values naturally depend on the height from which the vertical velocity and temperature data are obtained, as well as the averaging period used. Also, given the variability amongst the in-fire sensors, we are hesitant to rely on observations from a single (tower) location.

**_While the proposed method for estimating plume rise is somewhat novel, it is unclear why the authors don't use more observational data. The authors state that observations are limited and that is somewhat true, but given the recent publication of key wildfire plume datasets (RaDFIRE; Clements et al. 2018), the authors should really use wildfire observations verses low-intensity prescribed fires from RxCADRE. Another issue with the methodology presented in this study is that the authors use a vertical velocity scale for plume rise, but have no vertical velocity observations. Vertical velocity data are also available from the RxCADRE dataset. Additionally, a very recent paper by Rodriguez et al. (2020) show deep updraft velocities in a megafire that could be used as an extreme boundary for the parameterization. Additionally, a dataset from Lareau and Clements (2017) of a wildfire that includes plume evolution in a cross-wind is available as was also used by (Mallia et al. 2019)._**

Key issue with both RaDFIRE dataset and the one described by Rodriguez et al. (2020), is that to our knowledge neither provides spatiotemporally linked fire behavior data. Lareau and Clements (2017) use inverted Brigg's equations to produce a rough estimate of fire heat flux. In the absence of fire behavior observations, it is not possible to properly constrain our model using these data.

While in-situ tower data from RxCADRE indeed include vertical velocities over the passing fire front, the observations are limited to near-surface heights. Our parameterized vertical velocity scale is calculated

over the plume penetration region starting from upper ABL. We feel that comparing it with near-surface data may not be meaningful.

We were unable to locate a reference matching Mallia et al. (2019). We did find a paper by Mallia et al. (2018), but they've considered RxCADRE L2F dataset in their study.

***Some specific comments:***

***Line 148: It is not clear what the authors are defining as Fireline Intensity: ". . . fireline intensity parameter I, which is the kinematic heat flux into the atmosphere integrated across the fireline depth (in units of Km2s−1),. . ." I would call this the fire heat flux vs Byram's Fireline Intensity which has units of kW /m.***

We've added the following clarification in the manuscript:
…This velocity scale is related to the fireline intensity parameter $I$, which is the kinematic heat flux into the atmosphere integrated across the fireline depth (in units of $Km^2 s^{-1}$), and to the mixed-layer depth $z_i$. Note, that $I$ effectively corresponds to the kinematic form of Byram's Fireline Intensity (in units of $kW\ m^{-1}$).

We maintained the kinematic form throughout the manuscript to ensure unit consistency in the model equations.

***Line 205: Replace "lot" with "plot."***

Corrected.

***In Figure 2a, the mean plume centerline has a loop just downwind of the initial injection. Is this realistic? I would imagine that this feature represents the CBL, but would be averaged out as observed in the remainder of the downwind plume. Can the authors comment on this structure and whether this is realistic?***

Figure 2a shows cross-wind integrated, rather than spatially/temporally averaged view of the plume. Hence, we would expect some random oscillatory centerline behavior near the heat source, driven not only by CBL thermals, but also fluctuations in fire intensity and propagation speed (at prior time steps). These fluctuations are naturally suppressed further downwind, as the plume settles in the stable layers above the CBL. Plume widening further masks these oscillations in cross-wind view. We added the following clarification to the manuscript (2nd paragraph of Section 2.3):

…As a result, our approach is based on defining a region, where the concentration distribution is quasi-stationary. We consider the last frame of each simulation for this analysis. Using CWI integrated tracer values, we locate the plume centerline (Fig. 2a). Due to random effects of ABL thermals as well as fluctuations in fire intensity and propagation speed, both centerline height and concentration can vary near the heat source. These oscillations are naturally suppressed in the stable layers above the ABL, as the plume travels downwind and undergoes additional widening and mixing.

**References:**

Lareau, Neil P., and Craig B. Clements. "The mean and turbulent properties of a wildfire convective plume." *Journal of Applied Meteorology and Climatology* 56.8 (2017): 2289-2299.

Mallia, Derek V., et al. "Optimizing smoke and plume rise modeling approaches at local scales." *Atmosphere* 9.5 (2018): 166.

Rodriguez, B., et al. "Extreme Pyroconvective Updrafts During a Megafire." *Geophysical Research Letters* 47.18 (2020): e2020GL089001.